# Variation in albumin glycation rates in birds suggests resistance to relative hyperglycaemia rather than conformity to the pace of life syndrome hypothesis

Adrián Moreno Borrallo[1]*, Sarahi Jaramillo Ortiz[1,2], Christine Schaeffer-Reiss[1,2], Benoît Quintard[3], Benjamin Rey[4], Pierre Bize[5], Vincent A Viblanc[1], Thierry Boulinier[6], Olivier Chastel[7], Jorge S Gutiérrez[8], José A Masero[8], Fabrice Bertile[1,2†], Francois Criscuolo[1†]

[1]University of Strasbourg, CNRS, Institut Pluridisciplinaire Hubert Curien, Strasbourg, France; [2]National Proteomics Infrastructure, ProFi, Strasbourg, France; [3]Parc Zoologique et Botanique de Mulhouse, Mulhouse, France; [4]Lyon University 1, UMR CNRS 5558, Laboratoire de Biométrie et Biologie Evolutive, Villeurbanne, France; [5]Swiss Ornithological Institute, Sempach, Switzerland; [6]CEFE, Montpellier University, CNRS, EPHE, IRD, Montpellier, France; [7]Center of Biological Studies of Chizé (CEBC), UMR 7372 CNRS - La Rochelle University, Villiers-en-Bois, France; [8]Ecology in the Anthropocene, Associated Unit CSIC-UEX, Faculty of Sciences, University of Extremadura, Badajoz, Spain

*For correspondence:
adrian.moreno-borrallo@iphc.
cnrs.fr

†These authors contributed
equally to this work

Competing interest: The authors
declare that no competing
interests exist.

Reviewing Editor: Jenny
Tung, Max Planck Institute for
Evolutionary Anthropology,
Germany

## eLife Assessment

This **important** study uses extensive comparative analysis to examine the relationship between plasma glucose levels, albumin glycation levels, and diet and life history, within the framework of the 'pace of life syndrome' hypothesis. The evidence that glucose is positively correlated with glycation levels and lifespan is **convincing** and, although there are some limitations related to data collection, they likely make the statistically significant findings more conservative. As the first extensive comparative analysis of glycation rates, life history, and glucose levels in birds, the study has the potential to be of interest to evolutionary ecologists and the ageing research community more broadly.

**Abstract** The pace of life syndrome (POLS) hypothesis suggests that organisms' life history and physiological and behavioural traits should co-evolve. In this framework, how glycaemia (i.e. blood glucose levels) and its reaction with proteins and other compounds (i.e. glycation) covary with life history traits remain relatively under-investigated, despite the well-documented consequences of glucose and glycation on ageing, and therefore potentially on life history evolution. Birds are particularly relevant in this context given that they have the highest blood glucose levels within vertebrates and still higher mass-adjusted longevity compared to organisms with similar physiology as mammals. We thus performed a comparative analysis on glucose and albumin glycation rates of 88 bird species from 22 orders in relation to life history traits (body mass, clutch mass, maximum lifespan, and developmental time) and diet. Glucose levels correlated positively with albumin glycation rates in a non-linear fashion, suggesting resistance to glycation in species with higher glucose levels. Plasma glucose levels decreased with increasing body mass, but, contrary to what is predicted in the POLS hypothesis, glucose levels increased with maximum lifespan before reaching a plateau. Finally, terrestrial carnivores showed higher albumin glycation compared to omnivores

despite not showing higher glucose, which we discuss may be related to additional factors as differential antioxidant levels or dietary composition in terms of fibres or polyunsaturated fatty acids. These results increase our knowledge about the diversity of glycaemia and glycation patterns across birds, pointing towards the existence of glycation resistance mechanisms within comparatively high glycaemic birds.

## Introduction

The pace of life (POL) hypothesis (*Wikelski and Ricklefs, 2001*; *Ricklefs and Wikelski, 2002*; *Wikelski et al., 2003*) postulates that organisms' behavioural/physiological characteristics and life histories have co-evolved in answer to specific ecological conditions forming pace of life syndromes (POLS) along a fast-slow continuum (but see *Stott et al., 2024* for the more recent consideration of other axes). According to the classical approach from life history theory, slower organisms would have, for example, large body mass, late maturation, slow growth rate, small number of large offspring, and high longevity, with faster ones representing the opposite trait values (see *Stearns, 1989*; *Gaillard et al., 1989*; *Stearns, 1992*; *Vasilieva, 2022*) within this continuum. POL hypothesis added physiology to this continuum, with studies testing its predictions focusing often on metabolic rate (i.e. higher or lower metabolic rate corresponding to faster or slower POL, respectively; see e.g. *Trevelyan et al., 1990*; *Austad and Fischer, 1991*; *Speakman, 2005*; *de Magalhães et al., 2007*). However, key energy substrates related to metabolic performance, such as glucose concentrations in tissues, have been largely overlooked (but see *Kjeld and Ólafsson, 2008*; *Tomasek et al., 2019*; *Tomášek et al., 2022*; *Vágási et al., 2024*). Notably, glucose, a central energy source for many organisms, plays a pivotal role in metabolism, and there are some indications that its circulating blood or plasma levels correlate positively with metabolic rate per gram of mass at the interspecific level (*Kjeld and Ólafsson, 2008*; *Umminger, 1977*), and with whole-body metabolism at the intraspecific level (*Guo et al., 2023* for the case of non-diabetic humans).

Research focusing on plasma glucose becomes highly relevant as high glycaemia can entail costs that accelerate ageing. Along with other reducing sugars, glucose can react non-enzymatically with free amino groups of proteins, lipids, or nucleic acids, a reaction known as glycation (*Maillard, 1912*; *Suji and Sivakami, 2004*). This reaction, after several molecular reorganizations, can lead to the formation of a plethora of molecules called advanced glycation end-products (AGEs) (*Cerami et al., 1986*). AGEs can form aggregates in tissues that are well known to contribute to several age-related diseases, such as diabetes (e.g. reviews by *Poulsen et al., 2013*; *Chaudhuri et al., 2018*; *Khalid et al., 2022*; *Twarda-Clapa et al., 2022*). AGEs are also known to promote a proinflammatory state through their action on specific receptors called RAGEs (*Hofmann et al., 1999*; *Schmidt et al., 2001*).

Remarkably, birds show circulating glucose levels much higher than other vertebrate groups, on average almost twice as high as mammals (*Polakof et al., 2011*). These relatively high glucose levels might support the higher metabolic rates (*Lasiewski and Dawson, 1967*; *Speakman, 2005*) and body temperatures observed in birds compared to mammals (*King and Farner, 1961*; *Dawson and Hulbert, 1970*). In addition, elevated glycaemia is thought to be an adaptation to flight, providing birds with rapid access to an easily oxidizable fuel during intense bursts of aerobic exercise (see *Marsh et al., 2004*; *Weber, 2011*), such as during take-off and short-term flapping flights (*Braun and Sweazea, 2008*; *Szwergold and Miller, 2014a*). However, the presence of high glycaemia in birds is also paradoxical, given their remarkable longevity compared to their mammalian counterparts—living up to three times longer than mammals of equivalent body mass (*Lindstedt and Calder, 1976*; *Speakman, 2005*).

Several non-excluding hypotheses have been proposed to explain how birds apparently resist the pernicious effects of a high glycaemic state. One possibility is that birds might resist protein glycation through a lower exposure of lysine residues at the surface of their proteins (*Anthony-Regnitz et al., 2020*). Another hypothesis suggests that increased protein turnover may play a role (*Muramatsu, 1990*; *Makino and Kita, 2018*). Additionally, birds might benefit from more effective antioxidant defences (*Schweigert et al., 1991*; *Ku and Sohal, 1993*; *Hickey et al., 2012*) (although within Passeriformes [*Vágási et al., 2024*] shows no coevolution between glycaemia and antioxidant defences) or even the presence of 'glycation scavengers' (i. e. molecules that bind glucose avoiding it to react with proteins) such as polyamines which circulate at a high concentration (*Szwergold and Miller, 2014a*).

**eLife digest** Smaller animals often live shorter lives and use energy at a faster rate than their larger, longer-lived counterparts. This is partly related to differences in their resting metabolic rate, which is the energy expended to maintain basic bodily functions over a given time. For example, mice have high metabolic rates and short lifespans, whereas elephants live much longer and have lower metabolic rates per gram of body mass. However, many birds – despite having high metabolic rates – can live far longer than mammals of a similar size.

Birds also have the highest blood glucose levels of any vertebrate group. Through a process known as glycation, glucose and other sugars can attach themselves to molecules such as proteins. The resulting glycated proteins are thought to have negative effects on the body, which can contribute to the ageing process. Therefore, the amount of glycated protein in the blood is often used as a marker for harmful blood glucose levels in humans, which can be indicative of diseases such as diabetes. However, it remained unclear how birds resist the negative impacts of high blood glucose levels and glycation.

To investigate, Moreno-Borrallo et al. used measurements from 88 different bird species to explore how glucose levels and glycation of a protein called albumin are related to diet, lifespan and other variables. As expected, species with higher blood glucose levels had higher levels of albumin glycation. However, species with very high glucose levels showed relatively low glycation, suggesting these birds can resist the negative effects of high blood glucose.

Surprisingly, the analysis showed that species with higher glucose levels also tended to live longer, although this increase in lifespan eventually levelled off. This is contrary to the idea that species with higher metabolic activity have evolved shorter lifespans. Moreno-Borrallo et al. also showed that glucose levels decrease with body mass but are not related to any other traits. Glycation, on the other hand, is impacted by diet, with land (but not aquatic) carnivores showing higher levels than omnivores.

These analyses systematically explore how glucose and glycation levels relate to traits such as lifespan and diet across a wide range of bird species. The results will be valuable to evolutionary biologists and may also have implications for human health, particularly in understanding how glycation can be resisted during ageing. Future research should also focus on identifying which diets may help protect against glycation.

Moreover, it is generally considered that RAGEs are not present in avian tissues, suggesting possible adaptive mechanisms that would allow birds to avoid the inflammatory consequences associated with RAGE activation (*Eythrib, 2013*; *Szwergold and Miller, 2014b*). However, a putative candidate for RAGE and a case of AGE-induced inflammation in birds were identified by *Wein et al., 2020*, warranting further investigation on the occurrence of RAGEs in birds.

To date, although protein glycation levels have been measured in birds in several species, these results were mostly descriptive and concerned a small number of species (*Rendell et al., 1985*; *Miksik and Hodny, 1992*; *Andersson and Gustafsson, 1995*; *Beuchat and Chong, 1998*; *Ardia, 2006*; *Récapet et al., 2016*; *Ingram et al., 2017*; *Brun et al., 2022*; *Borger, 2024*). Additionally, many of these studies employ commercial kits that were not specifically designed for avian species, making it challenging to interpret the results (*Brun et al., 2022*). These limitations restrict our ability to draw general conclusions from these studies. Here, we performed a comparative analysis on primary data (for glycation and most of the glucose values, see 'Materials and methods' and Appendix 1) from 88 bird species belonging to 22 orders (see Appendix 1) and assessed whether and how bird glycaemia and glycation rates are linked to ecological and life history traits (including body mass). While doing this, we also checked whether glycation levels are influenced by circulating plasma glucose concentrations. Finally, as diet can influence glycaemia (see e.g. *Tomášek et al., 2022*; *Kapsetaki et al., 2023*; *Szarka and Lendvai, 2024* but *Tomasek et al., 2019* found no significant effects) and glycation, we included it in our analyses. We hypothesized that carnivorous birds would exhibit higher glycaemia and glycation levels than omnivorous or herbivorous species as diets with low carbohydrates, high proteins, and high fat are associated with comparatively high glycaemia and reduced insulin sensitivity in some vertebrates (e.g. *Coulson and Hernandez, 1983*; *Dobbs, 1985*; *Verbrugghe and Hesta, 2017*). Accordingly, a recent comparative study in birds showed higher blood glucose levels in

carnivorous species (*Szarka and Lendvai, 2024*). This may be attributed to high levels of constitutive gluconeogenesis, which has been confirmed in certain raptors (*Migliorini et al., 1973*; *Myers and Klasing, 1999*). Conversely, bird species with a high sugar intake, such as frugivores or nectarivores, are expected to exhibit high glycaemia, as observed in hummingbirds (*Beuchat and Chong, 1998*) and Passeriformes with such diet (*Tomášek et al., 2022*) (although the opposite was found in *Szarka and Lendvai, 2024*). In line with the POLS hypothesis, we also hypothesized that species with a slower POL should exhibit a lower glycaemia and higher glycation resistance (glycation levels for a given glycaemia) compared to species with a faster POL. Hence, we included the following four parameters in our analyses of glycaemia and glycation levels: body mass, maximum lifespan, clutch mass, and developmental time. Body mass is one of the main factors underlying life history, with higher body mass associated with 'slower' strategies and vice versa (*Stearns, 1983*). Previous studies have reported a negative relationship between body mass and glycaemia (*Umminger, 1975*; *Braun and Sweazea, 2008*; *Kjeld and Ólafsson, 2008*; *Tomasek et al., 2019*; *Szarka and Lendvai, 2024*) (but see *Beuchat and Chong, 1998*; *Scanes, 2016* with non-significant trends in birds, and *Tomášek et al., 2022* only for temperate species within Passeriformes). Therefore, we predicted that bird species that live longer, develop more slowly, and invest less per reproductive event (see Appendix 1 for further justification of the chosen variables) should show lower plasma glucose levels and albumin glycation rates (after controlling for body mass, phylogeny, diet, and glucose in the case of glycation; see 'Materials and methods').

## Materials and methods

### Species and sample collection

A total of 484 individuals from 88 species measured were included in this study (see *Appendix 1—table 1*). A detailed list with the provenance of the samples, including zoos, a laboratory population, and both designated captures and samples provided by collaborators from wild populations, is provided in *Supplementary file 3*, with a textual description in Appendix 1.

Blood samples were collected from the brachial or tarsal vein, or from the feet in the case of swifts, using heparin-lithium capillaries or Microvette (Sarstedt). As samples were collected mostly by different collaborators, handling times were not always recorded and could not be adjusted for, potentially rendering the results more conservative (for a more detailed discussion on potential stress effects on glucose, see Appendix 1), Samples were centrifuged at 4°C, 3500 × g for 10 min and plasma was aliquoted when a large volume was available. Subsequently, they were transported on dry ice to the Institut Pluridisciplinaire Hubert Curien (IPHC) in Strasbourg and stored at −80°C until analysis. Glycaemia was measured in the laboratory on the remaining plasma after taking an aliquot for glycation assessment using a Contour Plus glucometer (Ascensia Diabetes Solutions, Basel, Switzerland) and expressed in mg/dL. These of point-of-care devices have previously been assessed for its usage in birds (*Mohsenzadeh et al., 2015*; *Morales et al., 2020*), with several examples of its usage in recent literature related to the topic presented here (e.g. *Tomasek et al., 2019*; *Downs et al., 2010*; *Breuner et al., 2013*; *McGraw et al., 2020*). We also performed an assay with this particular brand on a subset of 46 samples from this study, coming from nine species distributed across the whole range of glucose values (three species with 'low' values, three with 'medium', and three with 'high', from both captive and wild populations, with five individuals per species, except one including six), confirming a positive linear correlation ($R^2_{marginal}$ = 0.66; $R^2_{conditional}$ = 0.84, for a model including the species as random factor) of this device with Randox GLUC-PAP colourimetry kits (p-value<0.001, unpublished data). Due to technical issues related to the incapability of the device to determine certain glucose values (not because of the glucose concentration, but perhaps the particular composition of the plasma samples from certain species), we could not determine glycaemia values of 95 individuals of those that we sampled and in which glycation levels were assessed (belonging to 40 species coming from different sources). For these species, if not a single individual had a glucose measurement (which was the case in 13 species), we obtained mean plasma glucose values reported for the species from the ZIMS database from Species360 (Species360 Zoological Information Management System [ZIMS] 2023, zims.Species360.org). This database provides plasma glucose data measured by colorimetric glucose kits on zoo specimens not necessary corresponding to those in which we measured glycation

values. The sample sizes for both glucose and glycation measurements, either from our measured individuals or from the ones from ZIMs, are reported in the *Supplementary file 5*.

## Glycation levels

Glycation levels for each individual were determined using liquid chromatography coupled to mass spectrometry (LC-MS), which is considered the gold standard for the assessment of protein glycation levels (see e.g. *Priego-Capote et al., 2014*) and has previously been used for birds (*Ingram et al., 2017*; *Brun et al., 2022*; *Zuck et al., 2017*). Given the relatively high time intrinsically taken by the employed methodology for analysing all the samples, linked to constraints in the access to the mass spectrometry devices, the whole set of samples of this study were analysed across several instances within a total timespan of less than 2 years (2021–2023). Only one sample from each individual was measured, given logistic limitations on the total number of samples that could be processed. Briefly, 3 µL of plasma were diluted with 22 µL of distilled water containing 0.1% of formic acid, followed by injection of 5 µL into the system. The glycation values used in the analyses represent the total percentage of glycated albumin, obtained by adding the percentages of singly and doubly glycated albumin. These percentages were calculated by dividing the areas of the peaks corresponding to each glycated molecule form (albumin plus one or two glucose) by the total albumin peak area (sum of the areas of glycated plus non-glycated molecules) observed in the spectrograms obtained from the mass spectrometry outcomes. These spectrograms represent the different intensities of signal for the components differentiated in the sample by their time of flight (TOF), which depends on the mass to charge ratio (m/z) of the ionized molecules. More detailed information regarding functioning of the method and data processing can be found in *Brun et al., 2022*. In cases where albumin glycation values dropped below the limit of detection, resulting in missing data, these individuals (10 individuals from four species) were excluded from the statistical analyses, as outlined below.

## Data on ecology and life history

Diets of individual species were extracted from the AVONET dataset (*Tobias et al., 2022*, coming from *Pigot et al., 2020*, and those adapted from *Wilman et al., 2014*) and life history traits from the Amniote Database (*Myhrvold et al., 2015*). After determining that AVONET diet classifications did not align with our research needs, minor changes were made after consulting the original *Wilman et al., 2014* database (see Appendix 1).

For missing data on life history traits after this stage, we extracted values, in order of preference, from the AnAge database (*Tacutu et al., 2018*) or the Birds of the World encyclopaedia (*Billerman et al., 2022*) by calculating mean values if an interval was given, and then averaging if male and female intervals were provided separately. For European species, maximum lifespan records have always been checked against the most recent Euring database (*Fransson et al., 2023*), and from *Chionis minor* and *Eudyptes chrysolophus* (not available from the previous sources) they were extracted from the Centre d'Etudes Biologiques de Chizé, Centre National de la Recherche Scientifique database. Several species from the zoo had maximum lifespan values available on the ZIMs from Species 360 (Species360 ZIMS 2023, zims.Species360.org), so these were also compared with the data we had from other sources.

In the case of variables (other than maximum lifespan) for which two different sources provided different records, an average value was calculated. For the maximum lifespan value, available sources were cross-checked and the highest value was always used. In the absence of satisfactory support from another source, values for maximum lifespan indicated as being anecdotal and of poor quality have been excluded from the analyses. A table with the adequate citations for each value is provided as part of the online available data (see 'Data availability' section). When the source is AVONET (*Tobias et al., 2022*), this database is cited, and when it is Amniote Database (*Myhrvold et al., 2015*), we cite the sources provided by them, so the references can be checked in *Myhrvold et al., 2015*. A list with all the additional references not coming from any of these databases or not provided by their authors (and that are not already in the main text) is given in *Supplementary file 4*.

## Statistical analyses

All analyses were performed in R v.4.3.2 (*R Development Core Team, 2023*). Alpha ($\alpha$)$\leq$0.05 was reported as significant, and 0.05$<\alpha\leq$0.1 as trends. General linear mixed models with a Bayesian

approach (*MCMCglmm* function in R; *Hadfield, 2010*) were performed. All models were run with 6 $\times$ $10^6$ iterations, with a thinning interval of 100 and a burn-in of 1000. The models were simplified by eliminating quadratic terms where they were not significant and selecting models with lower Akaike Information Criterion and Bayesian Information Criterion. Gaussian distribution of the response variables was assumed (after $\log_{10}$ conversion for glucose; see Appendix 1 for further discussions). The priors were established assuming variance proportions of 0.2 for the G matrix and 0.8 for the R matrix, with $\nu$ = 2. Less informative priors with equal variance proportions for each partition (residual and random) gave similar results. Lower $\nu$ values (0.1, 0.2, 0.002) were also tested, without success (i.e. simulations aborted before the established number of iterations). For the phylogenetically controlled analyses, consensus trees (each with the species included in the model) were obtained by using the *consensus.edges* function from the *phytools* package from R [*Revell, 2024*] from a total of 10,000 trees downloaded from Birdtree.org [*Jetz et al., 2012*] (option 'Hackett All species'; *Hackett et al., 2008*). For such purpose, a list of species with the names adapted to the synonyms available in such website was used (see *Supplementary file 5*), including the change of *Leucocarbo verrucosus* for *Phalacrocorax campbelli*, as the former was not available and it was the only species from the order Suliformes in our dataset, so that neither the position in the tree nor the branch length would be affected by this change. The consensus trees were included as *pedigree* in the models. We performed models with either glucose or glycation as dependent variables and with the diet, body mass, and life history traits as predicting variables (and glucose in the models with glycation as a response to assess glycation resistance; see 'Introduction'; another set of models without glucose were performed to test if there was a covariation of glycation itself, independently of glucose levels, covaried with life history; see Appendix 1). Generalized variance inflation factors (GVIFs) were calculated for all the models with more than one predictor variable to assess the collinearity of them as it may be expected for life history traits (see results in *Supplementary file 1*), considering values above 1.6 as slightly concerning, above 2.2 as moderately concerning, and above 3.2 as severely concerning (i.e. indicating high collinearity; following *Nahhas, 2024*). Models testing the effects of age and sex on glucose and glycation levels and the number of exposed lysine residues on glycation were also carried out. Finally, we performed models on glucose and glycation values controlling for the taxonomic orders included in the dataset (see *Supplementary file 5*). The effects of phylogeny on all models were determined by calculating the ratio of the variance estimated for the 'animal' variable, representing the *pedigree* (see Appendix 1) by the total variance (all random factors plus *units*). A thorough description of all of the models, including transformations of the variables and other details, is given in Appendix 1.

## Results

### Plasma glucose and albumin glycation variability across the birds' tree

Plasma glucose and albumin glycation values varied considerably across species (*Figure 1*). Considerable within-species repeatability (see Appendix 1) was observed (glucose: $R$ = 0.716, SE = 0.042, $CI_{95}$ = [0.619, 0.785], p-value<0.0001; glycation: $R$ = 0.703, SE = 0.042, $CI_{95}$ = [0.603, 0.768], p-value<0.0001).

We found significant differences between some orders for both glucose and albumin glycation. *Supplementary file 1* shows the raw outcomes of the models, showing such significant differences to the intercept and the credible intervals to perform the pairwise comparisons across all groups. To explore some of the details, we can see, for example, how Apodiformes showed the highest average glucose values (species average model: estimated mean = 351.8 mg/dL, $CI_{95}$[238.9, 514.2]; model with individuals: estimated mean = 349.8 mg/dL, $CI_{95}$[251.8, 476.7], n = 39 individuals, one species), followed by Passeriformes (species average model: estimated mean = 349.2 mg/dL, $CI_{95}$[279.6, 438.6]; model with individuals: estimated mean = 337.6 mg/dL, $CI_{95}$[274.5, 414.5], n = 84 individuals, eight species), while the Suliformes (species average model: estimated mean = 172 mg/dL, $CI_{95}$[117.6, 252.9]; model with individuals: estimated mean = 169.5 mg/dL, $CI_{95}$[120.9, 239.7], n=5 individuals, one species), Rheiformes (species average model: estimated mean = 173.7 mg/dL, $CI_{95}$[118, 254.7]; model with individuals: estimated mean = 172 mg/dL, $CI_{95}$[123.7, 239.8], n=9 individuals, one species), and Phoenicopteriformes (species average model: estimated mean = 176 mg/dL, $CI_{95}$[130, 241.1]; model with individuals: estimated mean = 141.2 mg/dL, $CI_{95}$[100.4, 198.2], n = 5 individuals, one species) show the lowest average values (for the species average models, the pattern is Suliformes

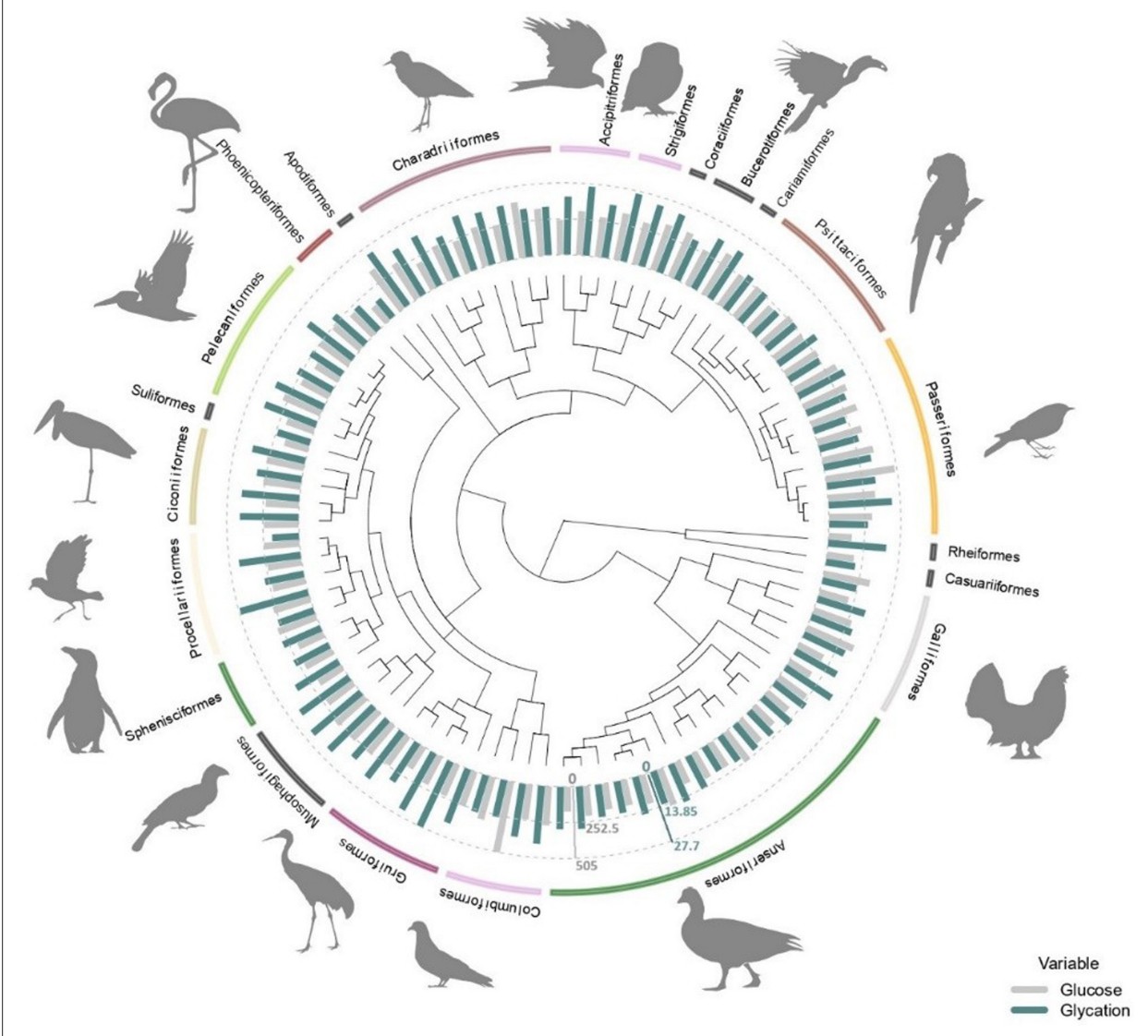

**Figure 1.** Average plasma glucose values in mg/dL (in grey) and average albumin glycation rate as a percentage of total albumin (in blue) from all the species used in this study (some of them with glucose values coming from ZIMs database; see 'Materials and methods') with the orders they belong to. Glucose and glycation values are standardized in order to be compared, with the dotted lines representing half the maximum and maximum values for each variable (as indicated by the axes in their corresponding colours), from inside out. Tree from a consensus on 10,000 trees obtained from 'Hackett All species' on Birdtree.org, including 88 species from 22 orders (see 'Materials and methods).

< Rheiformes < Phoenicopteriformes, while for the individual models is Phoenicopteriformes < Suliformes < Rheiformes).

For glycation, Strigiformes (species average model: estimated mean = 25.6 %, CI₉₅ **Suji and Sivakami, 2004**; **Dawson and Hulbert, 1970**; model with individuals: estimated mean = 25.3 %, CI₉₅[19.1, 31.5], n = 7 individuals, two species), Apodiformes (species average model: estimated mean = 25.5 %, CI₉₅[18.6, 32.4]; model with individuals: estimated mean = 24.8 %, CI₉₅[17.9, 32], n = 40 individuals, one species), and Coraciiformes (species average model: estimated mean = 24.5 %, CI₉₅[17.9, 31.6]; model with individuals: estimated mean = 23.9 %, CI₉₅[15.8, 32], n = 2 individuals, one species), all terrestrial carnivores as by our sampled species (see 'Discussion'), had the highest average.

On the other hand, Casuariiformes (species average model: estimated mean = 10.8 %, CI₉₅[3.7, 18.1]; model with individuals: estimated mean = 11.4 %, CI₉₅[2.7, 20.6], n=1 individuals, one species), Phoenicopteriformes (species average model: estimated mean = 11.2 %, CI₉₅[5.6, 16.7]; model with individuals: estimated mean = 10.5 %, CI₉₅[4.6, 16.3], n = 22 individuals, two species), and Suliformes

**Table 1.** Final glucose model with diet, body mass, life history traits, and sample provenance (wild versus captive; see Appendix 1) as explanatory variables, including the significant quadratic effect of maximum lifespan.

Posterior means, $CI_{95}$ and $p_{MCMC}$ from a phylogenetic MCMC GLMM model including n = 326 individuals of 58 species.Both glucose and body mass are $log_{10}$ transformed and life history traits are residuals from a phylogenetically controlled generalized least-squares model (pGLS) model of $log_{10}$ body mass and $log_{10}$ of the trait in question (see Appendix 1). Body mass was also centred to better explain the intercept, as 0 body mass would make no biological sense. The intercept corresponds to the omnivore diet, being used as the reference as it is considered the most diverse and 'neutral' group for this purpose. Significant predictors are indicated in bold.

| | Estimates | Lower 95% CI | Upper 95% CI | Sampling effort | $p_{MCMC}$ |
|---|---|---|---|---|---|
| **Intercept (omnivore)** | **2.387** | **2.268** | **2.5** | **59,900** | **<0.001** |
| Diet: carnivore terrestrial | 0.02 | –0.056 | 0.099 | 59,900 | 0.592 |
| Diet: aquatic predator | 0.034 | –0.047 | 0.112 | 59,900 | 0.393 |
| Diet: herbivore | –0.053 | –0.177 | 0.07 | 59,900 | 0.393 |
| Diet: frugivore/granivore | –0.055 | –0.239 | 0.135 | 59,900 | 0.558 |
| **Centred $Log_{10}$ body mass** | **–0.061** | **–0.106** | **–0.015** | **59,900** | **0.009** |
| Maximum lifespan | 0.107 | –0.035 | 0.253 | 59,900 | 0.142 |
| **Maximum lifespan$^2$** | **–0.616** | **–1.166** | **–0.095** | **59,900** | **0.026** |
| Clutch mass | –0.095 | –0.265 | 0.069 | 59,900 | 0.258 |
| Developmental time | 0.011 | –0.185 | 0.212 | 59,900 | 0.916 |
| Provenance: captive | 0.034 | –0.039 | 0.106 | 59,900 | 0.346 |

(species average model: estimated mean = 13.8 %, $CI_{95}$[7, 20.7]; model with individuals: estimated mean = 13.2 %, $CI_{95}$[5.6, 20.5], n = 5 individuals, one species) had the lowest average levels for the species average models, the pattern is Casuariiformes < Phoenicopteriformes < Suliformes, while for the individual models is Phoenicopteriformes < Casuariiformes < Suliformes. Graphs with raw data on species average and individual glucose and glycation values by order are shown in *Supplementary file 2* (*Supplementary file 2*). Estimates of phylogeny effects on the residuals of the models on glucose and glycation values differ if we consider the models with intraspecific variability or without it, but not so much between the models with and without life history traits (within the previous categories). In the case of species averages, the effect of the tree on residual glucose variation is lower and the estimation is less precise than for residual glycation variation, while for the models considering individual values, it is glycation residuals what shows lower levels of tree-related variance than glucose residuals (see *Supplementary file 1*).

## Bird glycaemia appears related to body mass and maximum lifespan

After controlling for intraspecific variation in our analyses, we found that the variations in glucose levels were significantly explained by the variations in body mass and both the linear and quadratic components of residual maximum lifespan (i.e. mass and phylogeny-adjusted maximum lifespan), but not by variations in diet (see below), clutch mass, and developmental time (see *Table 1* for the model including life history trait variables). Heavier species had lower glucose levels (see *Figure 2A*, drawn with the estimates from the model without life history traits, which includes more species). Glucose levels increase with increasing mass-adjusted lifespan until reaching a plateau (*Figure 2B*). Models that did not consider intraspecific variability show no significant effect of any of the aforementioned variables on glucose levels (see *Supplementary file 1*). The provenance of the samples (wild versus captive) only showed a trend to a higher glucose levels in the samples from captive individuals in the model without life history traits (estimate = 0.058; $CI_{95}$[–0.008, 0.125]; p-value = 0.083).

## Bird albumin glycation is related to glycaemia and diet

After controlling for intraspecific variability, we found that diet, contrary to what was observed for glucose (see *Figure 3A*, with predictions from the model without life history traits and *Supplementary file 2* with raw individual data from the same dataset), was affecting variation in albumin glycation

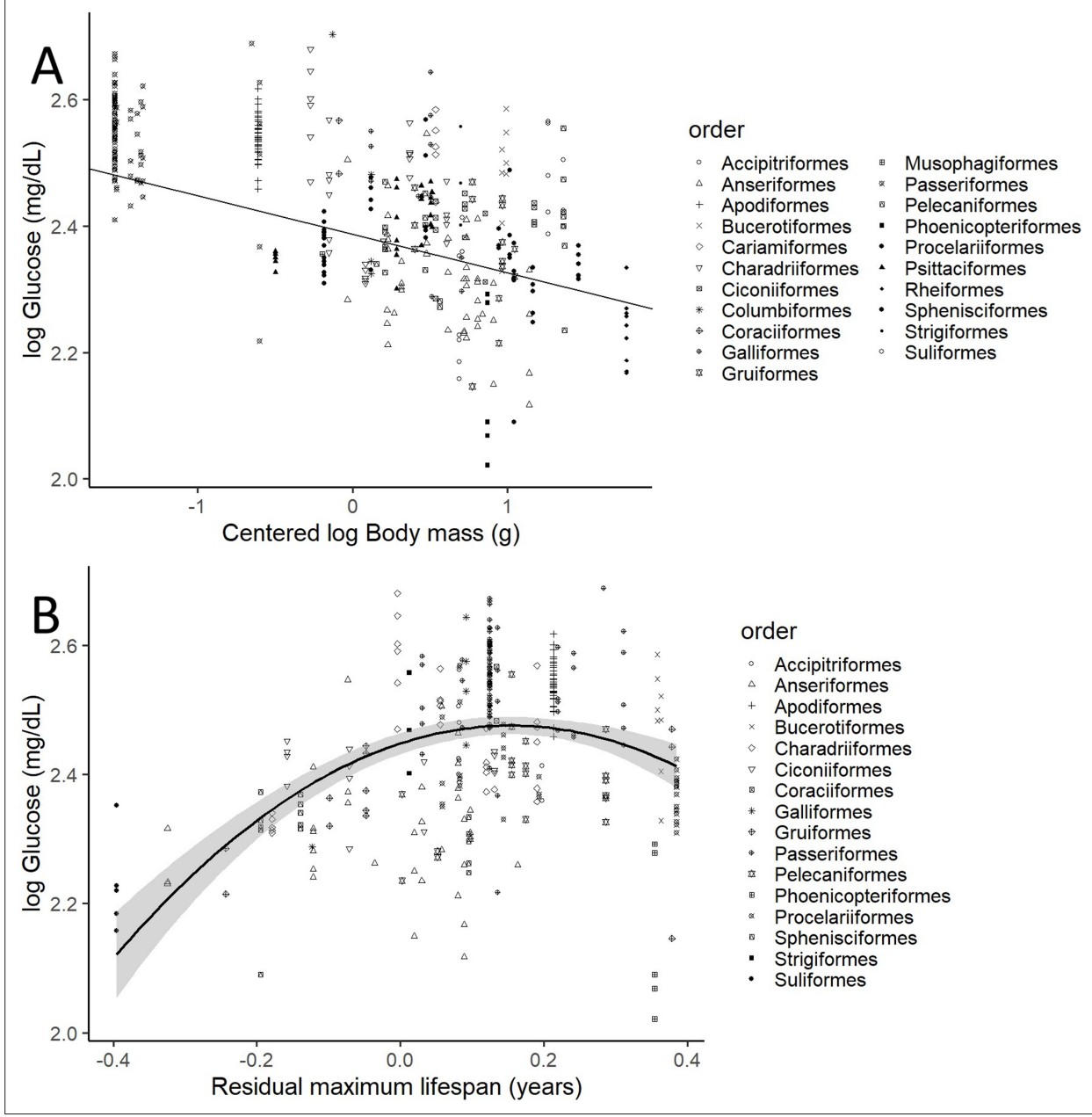

**Figure 2.** Plasma glucose levels (in mg/dL) variation as a function of (**A**) species mean-centred body mass and (**B**) residual maximum lifespan. Both glucose and body mass are log transformed. Maximum lifespan (in years) is given as the residues of a phylogenetically controlled generalized least-squares model (pGLS) model with body mass (in grams), both $\log_{10}$ transformed, so the effects of body mass on longevity are factored out (see Appendix 1). Different bird orders, are indicated by symbols, as specified on the legends at the right side of the graphs. (**A**) uses the values and estimates from the glucose model without life history traits (n = 389 individuals from 75 species), while (**B**) uses only the data points employed on the complete model (n = 326 individuals of 58 species).

rates, with terrestrial carnivorous species having higher glycation levels than omnivorous species (*Table 2* for complete model; see *Figure 3B* with predictions of the model without life history traits, and *Supplementary file 2* with raw individual data from the same dataset). However, for the models for species averages, there was only a trend on this pattern in the one including life history traits, and no effect in the other (see *Supplementary file 1*). The relation between glycation and glucose levels was positive and significant in all but the model that included life history traits but not intraspecific variation (see *Table 2* for the outcome of the model with individual values and life history traits and *Supplementary file 1* for the rest; see *Figure 4* with estimates from the model including individual

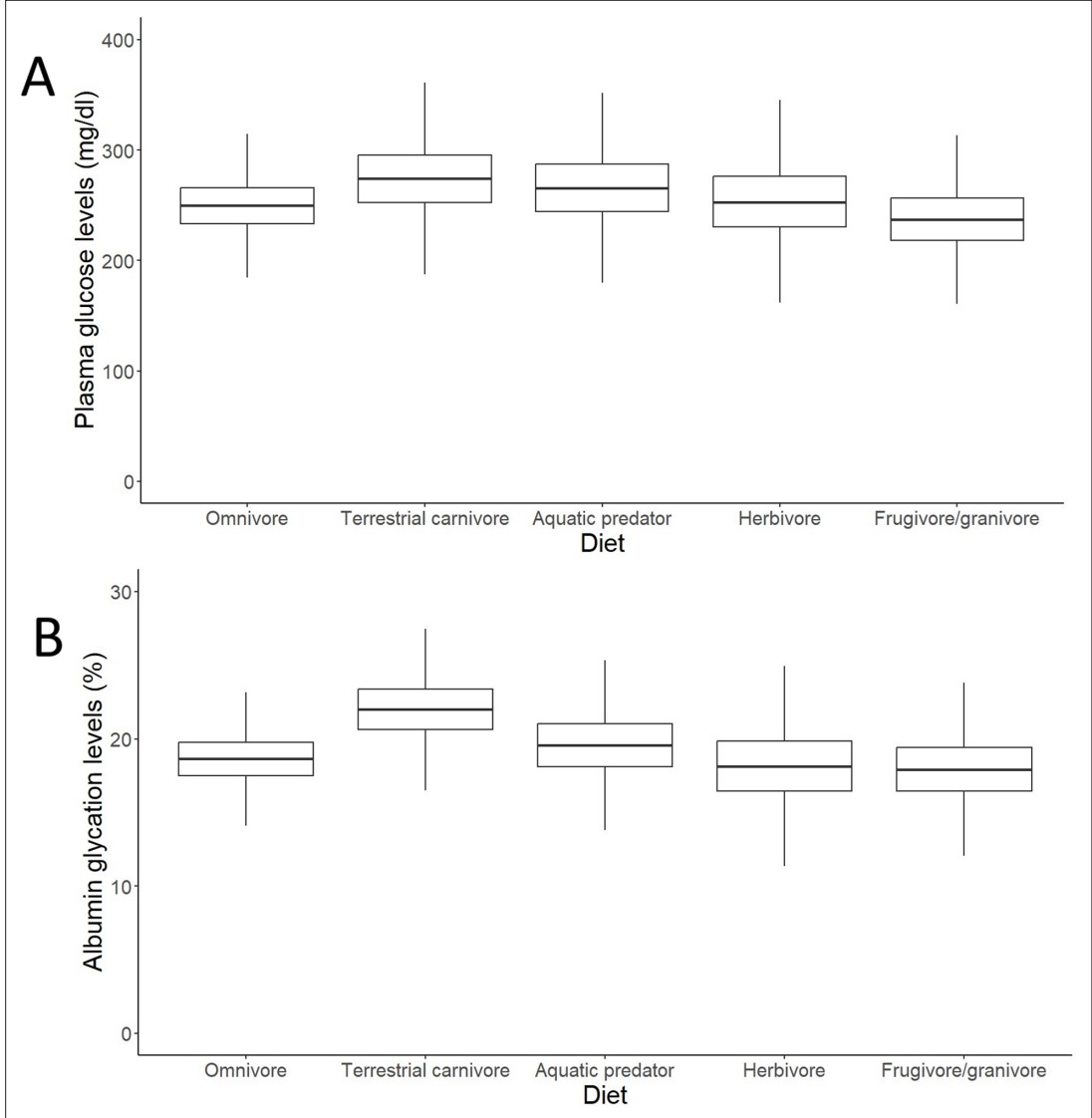

**Figure 3.** Outcomes of the models (estimates with interquartile ranges from the posterior distributions and whiskers representing credible intervals) on individual data on effects of diet on (**A**) plasma glucose levels and (**B**) albumin glycation in birds. Glucose levels are given in mg/dL, while glycation levels are a percentage of total plasma albumin which is found to be glycated. Terrestrial carnivores showed significantly higher glycation levels than omnivores (estimate = 21.62 %, CI₉₅[18, 25.95], $p_{MCMC}$ = 0.049). Models without life history traits, including more individuals, are represented, but the models with life history traits do not show differences in their qualitative predictions (i.e. higher albumin glycation in terrestrial carnivores than in omnivores; see *Supplementary file 1*).

variation but no life history traits, as it contains more species and the estimates are similar). Given the logarithmic relationship between glycation and glucose (see Appendix 1), the slope lower than one (see *Table 2*) implies that birds with higher glucose levels have relatively lower albumin glycation rates for their glucose, fact that we would be referring to as higher glycation resistance. The glycation models excluding glucose levels, and therefore testing for covariates of life history with glycation itself, without considering resistance, rendered similar results, with only the abovementioned dietary effects being significant (see *Supplementary file 1*).

Additional analyses looking at the number of exposed lysines in the albumin sequence of a species show no effect of this variable on albumin glycation rates ($\alpha$ = 10.39: CI₉₅ [–3.713, 24.993], $\beta$ = 0.246: CI₉₅[–0.144, 0.629], $P_{MCMC}$ = 0.196; *Supplementary file 1*).

We observed no significant effect of age relative to maximum lifespan nor sex on either glycaemia or glycation (see *Supplementary file 1*). In some models, GVIF for body mass and/or clutch mass were higher than 1.6, and in one case body mass is slightly above 2.2 (2.26, see *Supplementary file*

**Table 2.** Final glycation model with diet, body mass, glucose, and life history traits as explanatory variables.
Posterior means, $CI_{95}$, and $p_{MCMC}$ from a phylogenetic MCMC GLMM model including n = 316 individuals of 58 species. Glycation, glucose, and body mass are $\log_{10}$ transformed and life history traits are residuals from a linear model of $\log_{10}$ body mass and $\log_{10}$ of the trait in question (see Appendix 1). Body mass and glucose were also centred to better explain the intercept. The intercept corresponds to the omnivore diet, being used as the reference as it is considered the most diverse and 'neutral' group for this purpose. Significant predictors are indicated in bold, and the credible intervals are considered for making pairwise comparisons between the groups.

| | Estimates | Lower 95% CI | Upper 95% CI | Sampling effort | $p_{MCMC}$ |
|---|---|---|---|---|---|
| Intercept | **1.232** | **1.112** | **1.351** | 59,900 | **<0.001** |
| Diet: carnivore terrestrial | **0.101** | **0.017** | **0.187** | 59,900 | **0.021** |
| Diet: aquatic predator | 0.027 | −0.062 | 0.118 | 59,900 | 0.549 |
| Diet: herbivore | −0.019 | −0.161 | 0.119 | 59,936 | 0.781 |
| Diet: frugivore/granivore | 0.095 | −0.098 | 0.292 | 59,900 | 0.329 |
| Centred $\log_{10}$ body mass | 0.004 | −0.043 | 0.05 | 58,765 | 0.876 |
| $\log_{10}$ glucose | **0.137** | **0.012** | **0.255** | 59,900 | **0.027** |
| Maximum lifespan | 0.037 | −0.122 | 0.195 | 59,043 | 0.648 |
| Clutch mass | 0.151 | −0.03 | 0.346 | 59,900 | 0.114 |
| Developmental time | 0.04 | −0.188 | 0.266 | 59,303 | 0.725 |

1). This indicates that there may be moderate collinearity for this variables, but the fact that we used life history trait covariates that are excluding body mass associated variation, this small effects remain mysterious, but nevertheless probably not worrying.

## Discussion

### Plasma glucose level and albumin glycation rate (co)variation patterns suggest resistance mechanisms

Our findings show that glucose levels vary widely among different bird groups. Interspecific differences are partly explained by the allometric relationship of glycaemia with body mass (*Figure 2A*), which has already been reported in previous studies (*Szarka and Lendvai, 2024*; *Umminger, 1975*; *Braun and Sweazea, 2008*; *Kjeld and Ólafsson, 2008*; *Tomasek et al., 2019*). Indeed, Passeriformes, Apodiformes, and some Columbiformes (e.g. *Nesoenas mayeri* holds the highest value in our dataset) are found at the higher end of the glucose-level continuum, in accordance with their relatively small body mass and powered flight, while groups of larger birds such as Phoenicopteriformes, Anseriformes, Rheiformes, and Suliformes tend to show low glycaemia levels. This pattern is similar for glycation, with some groups of large birds (such as Phoenicopteriformes, Anseriformes, and Suliformes) showing the lowest levels of glycated albumin, while small birds (such as Apodiformes) had the highest values. Nevertheless, glycation remains high in some birds in relation to their glucose levels, as in Rheiformes, or low as in Psittaciformes or Passeriformes. The case of Procellariiformes, which are typically long-lived birds, is particularly striking, with some species exhibiting some of the highest glycation levels (*Calonectris diomedea* and *Macronectes giganteus*) and others some of the lowest (*Procellaria aequinoctialis*). This suggests that, if birds are protected against the deleterious effects of high glycaemia, certain species may have evolved mechanisms to efficiently prevent proteins to be glycated at high rates while others may have evolved mechanisms to resist the consequences of protein glycation. We should also bear in mind that some taxonomic groups may be under-represented in our study, or biases in species selection due to availability contingencies (e.g. species common in zoos or present in European countries) may exist, so further studies should target this under-represented groups in order to confirm our predictions.

As expected for a non-enzymatic reaction, just by the law of mass action, bird albumin glycation rates increase with plasma glucose concentration. However, the logarithmic nature of the relationship, and the fact that the slope is lower than one, suggests that species with higher plasma glucose levels

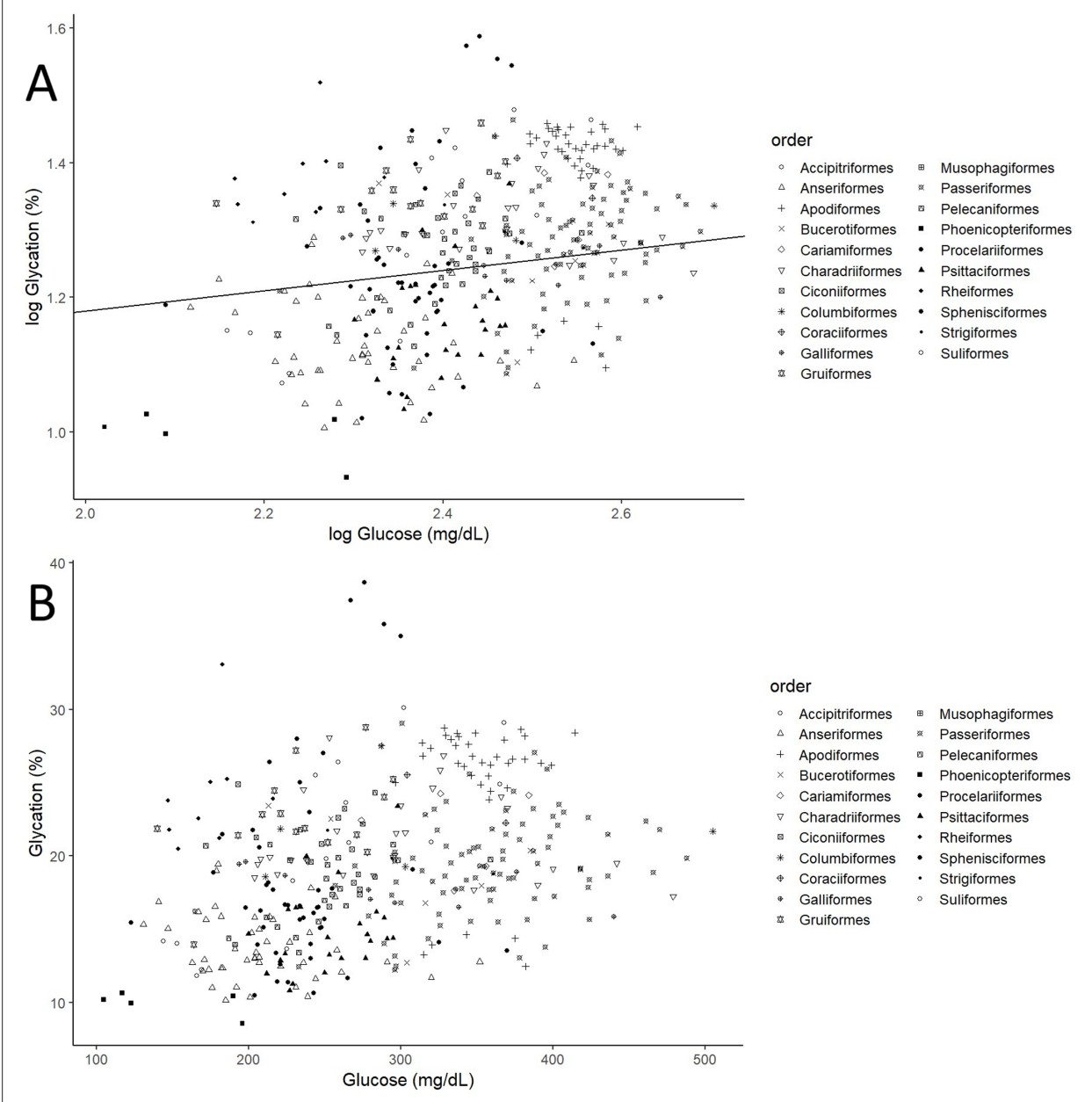

**Figure 4.** Individual albumin glycation rates (as a percentage of total albumin) variation as a function of individual plasma glucose values (mg/dL).
(**A**) Both variables $\log_{10}$ transformed, as in the model, including the line representing the predicted relationship. (**B**) Both variables in a linear form, to more explicitly illustrate the phenomenon referred to as higher albumin glycation resistance in birds with higher plasma glucose levels, inferred from the faster increase in glucose than albumin glycation, that is, the negative curvature of the relationship. Different bird orders are indicated by symbols, as specified on the legends at the right side of the graphs. The values and estimates used are from the glycation model without life history traits (n = 379 individuals from 75 species).

exhibit relatively greater resistance to glycation. This finding aligns with previous research indicating that in vitro glycation levels of chicken albumin increase at a slower rate than those of human albumin when glucose concentration is elevated (**Hackett et al., 2008**). Moreover, these levels are consistently lower than those of bovine serum albumin regardless of glucose concentration and exposition time (**Anthony-Regnitz et al., 2020**). As discussed in previous studies comparing chicken to bovine serum albumin (**Anthony-Regnitz et al., 2020**), or zebra finch to human haemoglobin (**Brun et al., 2022**), the lower glycation rates observed in bird proteins may result from a lower number of glycatable amino acids (e.g. lysines) in their sequence and/or their lesser exposure at the protein surface.

Our analyses do not succeed in indicating a significant positive relationship between average glycation levels and the number of glycatable lysines in the albumin sequence. This may be attributed to the limited number of species employed or the weak variation in the number of glycatable lysine residues, which are mostly ranging from 33 to 39. An interesting exception are the 18 glycatable lysine residues of flamingos (*Phoenicopterus ruber*), which also shows very low glycation levels (mean = 10.1%). However, the exceptions of 44 in zebra finches and 20 in godwits (*Limosa lapponica*, used in place of *Limosa limosa*) are nevertheless associated with very similar average glycation levels of 20.2% and 20.7% respectively.

## Plasma glucose relates with longevity and may be influenced by reproductive strategies

Our results were only in minor agreement with predictions from POLS theory: it holds for body mass, but not for the other life history variables tested. The only previous comparative study of glycaemia and life history traits to our knowledge (*Tomasek et al., 2019*) shows no relationship with mass-adjusted maximum lifespan in passerines. However, our study over 88 bird species on 22 orders revealed an increase in glucose with mass-adjusted longevity up to a plateau (see *Figure 2B*). Thus, the relationship between glucose and maximum lifespan may depend on differences between bird orders or be tied to specific species particularities not explored in our study. Such species particularities might involve additional undetermined ecological factors that modify the relationship of glycaemia with longevity. Further exploration of glucose metabolism in relation with lifestyle will bring further light on species-specific life history adaptations concerning glucose. For example, the species with the lower mass-adjusted maximum lifespan here was a cormorant (*L. verrucosus*), which have quite low glucose values for birds.

Regarding reproductive investment (i.e. clutch mass), our results show no relationship with glycation (see *Table 2*), while previous studies reported positive relationships with glycaemia in passerines (*Tomasek et al., 2019*; *Hackett et al., 2008*). Interestingly, most of the species with high clutch mass included in our study belong to the Anseriformes (*Supplementary file 2*, Figure ESM2.1). While these species exhibit very low glycaemia and albumin glycation rates, they are also characterized by a particular reproductive strategy compared to passerines, for whom clutch mass does not imply the same in terms of parental investment. For instance, Anseriformes, unlike Passeriformes, are precocial and their offspring are not very dependent on parental care. Furthermore, they are capital breeders, accumulating the energetic resources required for reproduction in advance rather than collecting them during reproduction. These species typically have large clutch sizes and incubation is usually carried out by females, who store internal fat resources to support this main part of the parental investment. In addition, ducklings are born highly developed, reducing the amount of care required post-hatching (see e.g. *Moreno, 1989*; *Winkler, 2016*). Consequently, their dependence on glucose as a rapid energy source for reproduction may be lower, with investment in this activity likely more closely linked to lipid accumulation. This could explain why we did not detect previously reported effects of clutch mass on glucose levels.

## Terrestrial carnivores show a paradoxically increased albumin glycation rate without increased plasma glucose levels

Contrary to our expectation of finding differences across dietary groups, plasma glucose did not significantly vary with species diet. This aligns with results previously reported by *Tomasek et al., 2019* for Passeriformes or by *Kapsetaki et al., 2023* for 160 species of vertebrates among which 48 were birds, but not with *Tomášek et al., 2022*, which showed glycaemia to be increased with the proportion of fruits/seeds in the diet, or with *Szarka and Lendvai, 2024*, which showed higher (mass-adjusted) glucose levels for terrestrial carnivores and insectivores (those included in our terrestrial carnivores' category) and lower for frugivorous/nectarivorous. On the other hand, intraspecific data indicates that changes in diet composition hardly affect glycaemia in birds (*Basile et al., 2022*). These studies suggest that glycaemia is tightly regulated independently of dietary composition within species, while it probably varies across species depending on the diet they are adapted to, although depending on the bird groups included in the analyses and the way of assessing it.

Although terrestrial carnivore species did not have significantly higher glycaemia levels in our study, they nonetheless demonstrated significantly higher albumin glycation rates, suggesting a potential

susceptibility to protein glycation. Besides unique structural features, this phenomenon could be due to lower albumin turnover rates in terrestrial carnivores. Studies in humans have linked higher oxidative damage to albumin with lower turnover rates (reviewed in *Wada et al., 2018*), which may extend to other species and post-translational modifications of albumin, such as bird's albumin glycation. However, the contrary outcome would be anticipated if protein intake were high as seen in carnivorous species (*Wada et al., 2018*; *Honma et al., 2017*).

Interestingly, the contrast between terrestrial and aquatic predators, which did not show such high glycation rates, suggests the beneficial influence of a substantial difference in food composition or the pre-eminence of another ecological factor, yet to be determined, which may explain lower glycation levels in aquatic predators. In terms of food composition, the proportion of essential n-3 PUFA (polyunsaturated fatty acids), particularly long-chain ones such as DHA and EPA compared with n-6 PUFA and short-chain n-3 PUFA, is different in aquatic and terrestrial environments (*Colombo et al., 2017*) and therefore in the diet of predators in each environment (e.g. *Koussoroplis et al., 2008*). For instance, low n-6 PUFA/n3-PUFA ratio have been shown to decrease insulin levels and improve insulin resistance in humans (*Li et al., 2019*). On the other hand, while some studies report that an increase in dietary n3-PUFA reduces the levels of glycated haemoglobin (HbA1c) in humans, this effect was not detected in many other studies and therefore remains inconclusive (reviewed in *Telle-Hansen et al., 2019*). The detailed study of the fat composition of the diets of terrestrial and aquatic predators, in particular the ratio between different types of PUFA, merits more attention. In addition, micronutrient content, such as circulating antioxidant defences, should also be taken into consideration. Indeed, a positive relationship between oxidative stress and glycated levels of bovine serum albumin has been reported (*Bavkar et al., 2019*). One hypothesis is that terrestrial predators have higher systemic oxidative stress levels compared to other species, which may be explained by defects in their antioxidant defences. Uric acid is one of the main non-enzymatic antioxidants in birds (*Klandorf et al., 1999*; *Machín et al., 2004*; *Stinefelt et al., 2005*), and uric acid levels are especially high in carnivore species that have a rich protein diet (*Machín et al., 2004*; *Harr, 2002*; *Smith et al., 2007*; *Cohen et al., 2009*; *Alan and McWilliams, 2013*). We should therefore take into account other important antioxidants such as vitamin E, vitamin A, and carotenoids, which may be less abundant in the diets of terrestrial carnivores (e.g. *Schneeberger et al., 2014*, but see *Ingram et al., 2017*). The question of whether the diet of aquatic carnivores provides a better intake of antioxidants would therefore requires a more detailed description of dietary habits. The herbivorous diet, meanwhile, despite the expected possibility contributing to lower glycaemia and glycation levels due to higher levels of PUFA compared to saturated fatty acids (SFA) (effects reviewed in *Telle-Hansen et al., 2019* for humans) and higher fibre content (see *Goff et al., 2018*), did not lead to significantly lower levels of either of these two parameters. However, the relatively low sample size in that group or the presence of outliers as *N. mayeri*, with the highest glucose levels of this dataset, makes the interpretation of the obtained results somehow limited. Therefore, further research should be carried out on these species to determine if the expected pattern would emerge with a better sampling.

Finally, differences between captive and wild populations could be considered as a source of variation in glucose or glycation levels due to a more sedentary lifestyle in captivity, with lower activity levels and a higher food intake likely to lead to increased circulating glucose levels in captive individuals. Additionally, differences in nutrient intake, such as antioxidants, specific fatty acids, or amino acids, between captive and wild populations could contribute to variations in glycation levels, as we saw above. Nevertheless, we think that this factor is unlikely to significantly affect our findings regarding diet categories because our study encompasses species from both captive and wild populations across various diet groups, particularly those exhibiting significant differences (e.g. omnivores versus terrestrial carnivores).

## Conclusion and perspectives

In conclusion, the avian plasma glucose levels measured here are generally higher or in the high range of values recorded for mammalian species (*Polakof et al., 2011*). Our study also concludes that there is considerable variation in plasma glucose levels and albumin glycation rates among bird species, with those with the highest glucose levels showing greater resistance to glycation. The correlation between plasma glucose and life history traits are primarily influenced by its inverse association with body mass, along with non-linear, mass-independent effects of longevity. Finally, although diet does not explain

plasma glucose levels in our study, terrestrial carnivores have higher albumin glycation rates than omnivores. Whether these intriguing results are explained by specific effects of certain dietary components, such as PUFAs and/or antioxidants, remains to be determined. Differences in plasma glucose levels, albumin glycation rates, and glycation resistance (as glycation levels adjusting for plasma glucose concentration) across bird species do not seem consistent with predictions according to the POL hypothesis, except for body mass. Further investigation is needed to elucidate the correlation between these traits and specific life conditions, such as reproductive strategy, migration patterns, flight mode, or more detailed diet composition. In addition, more in-depth exploration of glycation levels within high glycaemic groups such as the Passeriformes and other small birds, which make up a significant proportion of avian species, could provide valuable new insights. Similarly, investigating groups such as the Phoenicopteriformes or Anseriformes, which are at the other end of the glycaemic-glycation spectrum, could shed light on the origin of differences between avian orders. Furthermore, notable variations were observed between species within orders, as in the Procellariiformes, which with their high mass-adjusted longevity, and particularly given that some of the age-known individuals in our dataset (indeed, two specimens of snow petrels, *Pagodroma nivea*, showed ages higher than the maximum lifespan reported in the sources we explored, determining a new record) have still relatively low levels of glycation, may suggest that some species (the ones that do show higher glycation levels) are not intrinsically resistant to glycation, but rather to its adverse consequences for health. As the main limitations from this study, the usage of individuals from both wild and captive populations, sampled at different periods of the year, and the low sample size for some species, due to logistic constraints, may have introduced noise in the values reported that should be addressed in future studies by implementing a stricter sampling protocol. Also, a more thorough and accurate report and compilation of life history traits from multiple species would allow to increase the number of species included in this kind of analyses. Future research should also focus on specific species to unravel the physiological mechanisms mediating the effects of blood glucose and protein glycation on life history trade-offs, in particular mechanisms that may vary between taxa and contribute to characteristic adaptations in birds to mitigate the potentially negative effects of comparatively high glycaemia.

## Acknowledgements

This research was funded by an ANR (AGEs-ANR21-CE02-0009). We thank Charles-André Bost (CEBC CNRS) for providing data on some species. We would like to thank the SYLATR Association for collecting samples of wild Passeriformes as part of the MIGROUILLE program, and Manuela Forero and Frederic Angelier (CEBC CNRS) for providing some samples of certain Procellariiformes. We thank the French Parc des Oiseaux (Villars les Dombes), the bird keepers, and veterinarians at Mulhouse zoo for their contribution to the collection of blood samples and access to individual data from their captive birds. Data on seabirds from the French Southern Territories was collected within the framework of the ECONERGY (119), ECOPATH (1151), and ORNITHOECO (109) programs of the French Polar Institute (IPEV). These studies are part of the long-term Studies in Ecology and Evolution (SEE-Life) program of the CNRS. We are grateful to Mathilde Lejeune, Natacha Garcin, and Camille Lemonnier for their help in collecting those samples. We are thankful to Orsolya Vincze for producing *Figure 1*, Adrien Brown for determining the number of lysines exposed in albumin sequences, Claire Saraux for her help with the statistics, and F Stephen Dobson for reading and commenting on the manuscript. Finally, we are thankful to Ascensia Diabetes care for their generous donation of glucometers and strips for glucose measurement. All authors gave their approval for publication.

# Additional information

## Funding

| Funder | Grant reference number | Author |
|---|---|---|
| Agence Nationale de la Recherche | AGEs - ANR21-CE02-0009 | Adrián Moreno Borrallo<br>Sarahi Jaramillo Ortiz<br>Christine Schaeffer-Reiss<br>Fabrice Bertile<br>Francois Criscuolo |

The funders had no role in study design, data collection and interpretation, or the decision to submit the work for publication.

## Author contributions

Adrián Moreno Borrallo, Conceptualization, Data curation, Formal analysis, Funding acquisition, Investigation, Visualization, Methodology, Writing – original draft, Writing – review and editing, A. Moreno-Borrallo contributed to the development of the questions, gathered the diet and life-history data, performed the statistical analyses, participated in the glucose measurements and validation of the methodology for it, and led the writing; Sarahi Jaramillo Ortiz, Methodology, Writing – review and editing, S. Jaramillo Ortiz performed the mass spectrometry analyses for protein glycation measurements, contributed to the glucose measures and commented on the manuscript; Christine Schaeffer-Reiss, Supervision, Methodology, Project administration, C. Schaeffer supervised the mass spectrometry analyses for protein glycation measurements; Benoît Quintard, Resources, Methodology, Writing – review and editing, B. Quintard leads the health monitoring of Mulhouse zoo bird collection, organized and realized most of Mulhouse zoo samplings and commented on the manuscript; Benjamin Rey, Resources, Writing – review and editing, B. Rey contributed to the collection of samples from Parc des Oiseaux (Villars-les-Dombes, France) and commented on the manuscript, Methodology; Pierre Bize, Resources, Methodology, Writing – review and editing, P. Bize leads the monitoring of Alpine swift populations from which the samples were obtained, he helped collecting them and commented on the manuscript, Funding acquisition; Vincent A Viblanc, Resources, Writing – review and editing, T. Boulinier. and V. A. Viblanc coordinated the ECONERGY and ECOPATH polar programs, organised the collection of samples on subantarctic seabirds, and commented on the manuscript, Funding acquisition, Methodology; Thierry Boulinier, Resources, Writing – review and editing, T. Boulinier. and V. A. Viblanc coordinated the ECONERGY and ECOPATH polar programs, organised the collection of samples on subantarctic seabirds, and commented on the manuscript, Funding acquisition, Methodology; Olivier Chastel, Resources, Olivier Chastel collaborated on collecting marine bird samples and commented on the manuscript, Funding acquisition, Methodology; Jorge S Gutiérrez, Resources, Software, Writing – review and editing, J. S. Gutiérrez and J. A. Masero contributed with samples from Spain and commented on the manuscript. J. S. Gutiérrez helped with part of the statistic scripts and methodology, Funding acquisition, Investigation; José A Masero, Resources, Writing – review and editing, J. S Gutiérrez and J. A. Masero contributed with samples from Spain and commented on the manuscript; Fabrice Bertile, Conceptualization, Supervision, Funding acquisition, Validation, Project administration, Writing – review and editing, F. Criscuolo and F. Bertile conceived the idea, directed most of the sample collection and logistics and contributed significantly to the writing; Francois Criscuolo, Conceptualization, Supervision, Funding acquisition, Validation, Project administration, Writing – review and editing, F. Criscuolo and Fabrice Bertile conceived the idea, directed most of the sample collection and logistics and contributed significantly to the writing

## Author ORCIDs

Adrián Moreno Borrallo ⓘ https://orcid.org/0000-0002-2924-1153
Sarahi Jaramillo Ortiz ⓘ http://orcid.org/0000-0002-9153-4205
Christine Schaeffer-Reiss ⓘ http://orcid.org/0000-0003-0672-1979
Benjamin Rey ⓘ http://orcid.org/0000-0002-0464-5573
Pierre Bize ⓘ https://orcid.org/0000-0002-6759-4371
Vincent A Viblanc ⓘ http://orcid.org/0000-0002-4953-659X
Thierry Boulinier ⓘ http://orcid.org/0000-0002-5898-7667
Jorge S Gutiérrez ⓘ https://orcid.org/0000-0001-8459-3162

José A Masero http://orcid.org/0000-0001-5318-4833
Fabrice Bertile http://orcid.org/0000-0001-5510-4868
Francois Criscuolo http://orcid.org/0000-0001-8997-8184

## Ethics

This study followed all the legal considerations, with the ethic authorisations from the French Ministry of Secondary Education and Research, n°32475 for the zebra finches sampling, the Swiss Veterinary Office (FSVO) n° 34497 for the Alpine swifts, the Ethics Committee of the University of Extremadura (licenses112//2020 and 202//2020) and the Government of Extremadura (licenses CN0012/22/ACA and CN0063/21/ACA) for the godwits and terns from Spain, and Sampling in Terres Australes et Antarctic Françaises was approved by a Regional Animal Experimentation Ethical Committee (French Ministry of Secondary Education and Research permit APAFIS #31773-2019112519421390 v4 and APAFIS#16465-2018080111195526 v4) and by the Comité de l'Environnement Polaire and CNPN (A-2021-55). The samples from the Mulhouse zoo were taken by its licensed veterinary with capacity number: 2020-247-SPAE-162.

Reviewer #2 (Public review): https://doi.org/10.7554/eLife.103205.4.sa1
Author response https://doi.org/10.7554/eLife.103205.4.sa2

---

# Additional files

## Supplementary files

MDAR checklist

Supplementary file 1. Detailed models' outcomes.

Supplementary file 2. Additional figures.

Supplementary file 3. Provenance of the samples.

Supplementary file 4. References for the data values used as quoted in the dataset (*Supplementary file 5*).

Supplementary file 5. Dataset.

Supplementary file 6. Code.

## Data availability

All data used, including references for the values taken from bibliography or databases, are available as Electronic Supplementary Material (*Supplementary file 5*) and on figshare at https://doi.org/10.6084/m9.figshare.28669523.v1. Code is made available as Electronic Supplementary Material (*Supplementary file 6*).

The following dataset was generated:

| Author(s) | Year | Dataset title | Dataset URL | Database and Identifier |
|---|---|---|---|---|
| Moreno-Borrallo A, Jaramillo-Ortiz J, Schaeffer-Reiss C, Quintard B, Rey B, Bize P, Viblanc VA, Boulinier T, Chastel O, Gutiérrez JS, Masero JA, Bertile F, Criscuolo F | 2025 | Data from: Variation in albumin glycation rates in birds suggests resistance to relative hyperglycaemia rather than conformity to the pace of life syndrome hypothesis | https://doi.org/10.6084/m9.figshare.28669523.v1 | figshare, 10.6084/m9.figshare.28669523.v1 |

The following previously published datasets were used:

| Author(s) | Year | Dataset title | Dataset URL | Database and Identifier |
|---|---|---|---|---|
| Myhrvold NP, Baldridge E, Chan B, Sivam D, Freeman DL, Ernest SKM | 2025 | An amniote life-history database to perform comparative analyses with birds, mammals, and reptiles | https://doi.org/10.6084/m9.figshare.c.3308127.v1 | figshare, 10.6084/m9.figshare.c.3308127.v1 |

*Continued on next page*

*Continued*

| Author(s) | Year | Dataset title | Dataset URL | Database and Identifier |
|---|---|---|---|---|
| Tobias JA, Sheard C, Pigot AL, Devenish AJM, Yang J, Sayol F, Neate-Clegg MHC, Alioravainen N, Weeks TL, Barber RA, Walkden PA, MacGregor HEA, Jones SEJ, Vincent C, Phillips AG, Marples NM, Montaño-Centellas FA, Leandro-Silva V, Claramunt S, Darski B, Freeman BG, Bregman TP, Cooney CR, Hughes EG, Capp EJR, Varley ZK, Friedman NR, Korntheuer H, Corrales-Vargas R, Trisos CH, Weeks BC, Hanz DM, Töpfer T, Bravo GA, Remeš V, Nowak L, Carneiro LS, Amilkar J, Moncada R, Matysioková B, Baldassarre DT, Martínez-Salinas A, Wolfe JA, Chapman PM, Daly BO, Sorensen MC, Neu A, Ford MA, Mayhew RJ, Silveira LF, Kelly DJ, Annorbah NND, Pollock HS, Grabowska-Zhang AM, McEntee JP, Gonzalez JGT, Meneses CG, Muñoz MC, Powell LL, Jamie GA, Matthews TJ, Johnson O, Brito GRR, Zyskowski K, Crates R, Harvey MG, Zevallos MJ, Hosner PA, Bradfer-Lawrence T, Maley JM, Stiles FG, Lima HS, Provost KL, Chibesa M, Mashao M, Howard JT, Mlamba E, Chua MAH, Li B, Isabel Gómez M, García NC, Päckert M, Fuchs J, Ali JR, Derryberry EP, Carlson ML, Urriza RC, Brzeski HE, Prawiradilaga DM, Rayner MJ, MIller ET, Bowie RCK, Lafontaine RM, Scofield RP, Lou Y, Somarathna L, Lepage D, Illlif M, Neuschulz EL, Templin M, Dehling DM, Cooper JC, Pauwels OSG, Analuddin K, Fjeldså J, Sweet PR, Naka LN, Brawn JD, Aleixo A, Böhning-Gaese K, Rahbek C, Fritz SA, Thomas OH, Schleuning M | 2022 | AVONET: morphological, ecological and geographical data for all birds | https://figshare.com/s/b990722d72a26b5bfead | figshare, b990722d72a26b5bfead |

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

## Appendix 1

### Study species

Of all the species on the study, 65 came from captive populations of the Mulhouse zoo (Mulhouse, France) and the Parc des Oiseaux (Villars-les-Dombes, France). These samples were collected during annual vaccination campaigns in October 2018 and 2021. Zebra finch (*Taeniopygia guttata*) samples were taken in August–September 2022 from individuals kept in captivity in the animal facilities of the Department of Ecology, Physiology and Ethology, Institut Pluridisciplinaire Hubert Curien, Strasbourg, France. Alpine swift (*Tachymarptis melba*) samples were collected in May and August 2023 from wild individuals breeding in colonies in Switzerland (see e.g. *Bize et al., 2006*; *Bize et al., 2008*; *Bize et al., 2014*). Adult black-tailed godwits (*Limosa limosa*) were sampled in February 2022 in Extremadura rice fields, southwestern Spain. Adult gull-billed terns (*Gelochelidon nilotica*) were sampled in May–June 2022 from a breeding colony located at Villalba de los Barros reservoir in Extremadura, southwestern Spain. All passerine samples, excluding zebra finches and Bali myna (*Leucopsar rothschildii*), were obtained from wild individuals captured using mist nets during a migration monitoring project in a wetland at Sainte-Soline (France) in August 2022. Scopoli's shearwater (*C. diomedea*) samples were collected in 2011 in the Chafarinas Islands, Spain. The remaining samples from Procellariiformes species, penguins (Sphenisciformes), cormorants (Suliformes), and all marine Charadriiformes were collected from wild populations in the French Southern Territories of Crozet and Kerguelen during 2022. Sample sizes for each species measured by our team went from 1 to 60 for glucose (mean = 5.32; σ = 7.07; median = 4.5; mode = 5) and from 1 to 55 for glycation (mean = 5.19; σ = 7.9; median = 4; mode = 5). Excepting zebra finches (*Taeniopygia guttata*) (60 for glucose, 55 for glycation) and Alpine swifts (*Tachymarptis melba*) (39 for glucose, 40 for glycation) whose sample sizes are bigger because these measurements were originally collected on untreated animals for separate analyses (unpublished data), the values are lower and vary much less in number (1–16 for both glucose and glycation), and are limited in size by logistical reasons, mainly related to mass spectrometry operation. The values reported here for glycation are counting the individuals included in the statistical analyses, not the ones sampled, as some individuals had glycation values under the limit of detection, that were eliminated from the analyses, as stated in the main text. However, when considering glucose values coming from the species from ZIMs database, the sample sizes were bigger 17 387 individuals (mean = 90.46; σ = 26.31; median = 55; see main text). The validity of the glucose and glycation values from our dataset to represent the species (as sample size per species was often low) was assessed by performing general linear mixed models (GLMM; lmer() function from lme4 package in R; *Bates et al., 2015*) with either glucose or glycation as dependent variables and 'species' as a random factor for the intercept only, and then obtaining the repeatability score with rptGaussian() from rptR package (*Nakagawa and Schielzeth, 2010*) in order to determine if the variability between species is higher than within species.

### Possible effects of stress

Although glucose levels are known to be affected by the stress response, and thus by handling time (see e.g. *Remage-Healey and Romero, 2000*; *Viblanc et al., 2018*; *Ryan et al., 2023* for birds), and time between bird capture and blood sampling ('handling time') is not measured here, there are some reasons to consider that our results are still robust. First, most individuals were sampled shortly after captured, so that the stress response could not play a big role on glucose variation. Second, we considered that this change in glucose levels with stress, at least in certain species, may be driven more by an increase in variation than by an increase in average values, which means that our study would remain conservative, so that the probability of finding significant results is lower, but the patterns of variation that we did find are more robust. We base this idea on an analysis we did on publicly available data from 160 species of Passeriformes from *Tomášek et al., 2022*, in which we performed a LMM on glucose with sampling time in min (0, 15, and 30) as fixed factor (see *Supplementary file 1*), and individual nested within species as random factor, showing a clear heteroskedasticity across sampling times, and a model without the fixed factor. This way, we also compared the percentages of variance explained by the species and individual when considering all reported glucose values (i.e. at all sample times) together with the variance when the time effect was accounted for (see *Supplementary file 1*), showing a lower species repeatability when sampling time is not considered. Moreover, their species-explained variance when sampling

time is not considered is lower than in our study, probably in relation to our higher taxonomic cover. Given the mentioned heteroskedasticity of the data, a Kruskal–Wallis test (non-parametric) was performed to explore the differences in glucose values across sampling times, reporting a significant result (see *Supplementary file 1*), which indicates differences across the times, that are nevertheless not necessarily associated with different means. Third, preliminary analyses performed by our team on data from Alpine swifts and zebra finches (unpublished results), in which many individuals were sampled during the same session, and therefore differences in stress levels across individuals may be present, show no significant effect of time on glucose variation, particularly when the individuals were overnight fasted, as it is the case for zebra finches

## Diet data clarifications

In many cases, adjustments were necessary because the criteria used by *Pigot et al., 2020* to classify species (subsequently used in AVONET) were based on the ecological niche and its relationship with morphology rather than considering the potential association of dietary composition with physiological variables. To address this limitation, we used the 'Trophic.Niche' variable from AVONET and merged some categories: 'Vertivore', 'Invertivore', and 'Scavenger' into 'Carnivore terrestrial', 'Frugivore', and 'Granivore' into 'Frugivore/granivore' and 'Herbivore terrestrial' and 'Herbivore aquatic' into 'Herbivore', due to lack of sufficient species on each separate category (as in 'Herbivore terrestrial' and 'Herbivore aquatic') and sometimes on difficulties to tease apart the diet categories for our species (as in 'Frugivore' and 'Granivore', as most of our species classified within these groups had a 50%-50% or similar distribution in their food source between both categories, zebra finches being the only exception being exclusively granivorous). Furthermore, some considered in AVONET as 'omnivores' were reclassified, and only considered 'omnivores' if diet was widely distributed between animal and plant sources, and not only within animal or plant categories (e.g. a species that eats invertebrates, vertebrates, and carrion was not considered by us as an 'omnivore', but as either an aaquatic predator" or a 'carnivore terrestrial', depending on its main feeding source).

## Statistics

### Life history traits

Given that all the three life history parameters (maximum lifespan, clutch mass, and developmental time) used in our analyses are widely recognized to be strongly correlated with body mass, and interpretations of the variation in life history traits can change if body mass is not adjusted for (*Jeschke and Kokko, 2009*). we chose to adjust the life history traits by body mass prior to run our analyses, as done by *Tomasek et al., 2019*, the most similar study on glucose levels in birds in function of life history, so that our results are easily comparable. This was done by log10 transforming all variables and performing a phylogenetically controlled generalized least-squares model (pGLS, gls function in R; *Pinheiro and Bates, 2000*) including a correction for phylogeny (assuming a Brownian model and the same tree as described in the main text for the MCMC GLMMs), from which the residuals were extracted. These residuals were the variables used in the main models when referring to 'life history traits', and they are also the same thing as when referring to mass-adjusted variables across the text. Maximum lifespan was used here as a proxy for longevity. Despite concerns about its use (e.g. *Krementz et al., 1989*; *Ronget and Gaillard, 2020*), it remains the most widely available longevity-related data in many databases for a large number of species. Clutch mass, calculated by multiplying clutch size by egg mass (as in *Tomasek et al., 2019*), it was used as an indicator of reproductive investment. Finally, developmental time (incubation time plus number of days to fledging) was used as a proxy for growth, given that growth rate was difficult to calculate due to lack of information on fledging mass. These variables were also chosen as they were among the ones available for a greater number of species from the sources we used, and for comparative purposes with *Tomasek et al., 2019*. Including more variables as, for example, number of clutches per year (one of the other available variables for a considerable number of species) would have further reduced the sample size without, in our opinion, adding much more information, as reproductive effort is already accounted for in the clutch mass variable.

### Main models

Models were performed in the following way (see *Appendix 1—table 1*): first, in order to account for all the species in the dataset (as individual values were not available for all the species, with no individual at all in some cases; see 'Species and sample collection' section from main text)

with average glucose and then with average glycation as dependent variables, and later with the individual values in another set of models. Body mass and diet were always included as independent variables. All of this was repeated also including life history traits but with the 66 species for which data were available (only species for which we had all the selected life history variables were kept). Glycaemia and body mass were log10 transformed and centred in all the models (centring of glucose only occurred when it was a covariable) in order to better interpret the intercepts (*Schielzeth, 2010*). In models predicting glycation, plasma glucose was also added as a covariate. For these particular models, linear (glycation – centred glucose) and logarithmic (log$_{10}$Glycation – log$_{10}$(centred glucose) relationships were assessed, comparing the deviance information criteria) of the different models, log–log relationship showing much lower DIC than the linear relationship. Semilogarithmic relationships (both glycation – log$_{10}$(centred glucose) and log$_{10}$ glycation – centred glucose) were also assessed, but DICs were similar between these and either the linear or the log–log model, with much lower DIC for the ones using log$_{10}$glycation as dependent variable. Also, all these models showed similar results regarding the significant effects found, and if glucose and glycation were standardized, the estimates of the slopes were very similar for all of them. Therefore, the log–log model was chosen. Identical models without including glucose as a covariate were also performed (for the glycation models including life history traits) to determine if glycation itself (not glycation resistance, as defined in the main text), showed a covariation with life history. Quadratic component of maximum longevity was also included in glucose and glycation models including life history data, after data exploration suggested such pattern. The potential effects of captivity were also tested by a factor with two levels indicating the provenance of the samples (captive or wild), in preliminary analyses with a lower number of samplings (104 iterations, burnin=1000). As no effects were found for this variable in the glycation models, it was only maintained in the glucose models thereafter. Besides, in glucose models, the methodology used for determining it (glucometer or kit) was included as a random factor, in order to control for a potential bias in the results. The species was added as a random factor (given a tree provided within the pedigree argument of the MCMCglmm function).

Since glucose values were not available for all individuals, our models addressing intraspecific variability were limited to 379 individuals from 75 species when considering only diet and body mass predictors (and glucose when modelling glycation). Similarly, only 316 individuals from 58 species were incorporated into the models when life history traits were introduced. The species identity was always included as a random factor twice: one to control for the phylogeny and the other to avoid the pseudo-replication effect of having repeated measures within species. The method used to determine plasma glucose levels was not included in this case since here the glucometer was used in all cases.

Gaussian distribution was always assumed for the models, as even if glycation values are, strictly speaking, proportions, not only they follow a continuous distribution (as their calculation come from the division of area values that vary in a continuous way; see main text), but they vary within a limited range (8.57-38.65%, mean = 18.8 %, $\sigma$ = 5.3) never sufficiently close to 0 or 1 and also do not represent probabilities given by observations of cases as to be treated as binomial with log link function. Furthermore, we believe that there are some reasons why logit transformation, as previously also recommended for proportional data (*Warton and Hui, 2011*), is not the best option in this case: first, logit calculations are suitable for exploring how probabilities of certain outcomes differ between groups, which is not the case in our analyses; second, exploration of the data suggested this transformation did not markedly alter the variance structure of the variable, which was close to a Gaussian curve, while making more difficult the interpretation of the outcomes of the models.

Model convergence and lack of autocorrelation was checked by visual inspection of trace of the model MCMC simulations (see trace and density plots of posteriors in *Supplementary file 1*).

## Models on age and sex, on lysine residues, and on orders

We performed supplementary analyses only with relative age (when it was known, in most cases for birds born in captivity, or belonging to wild populations with demographic monitoring), including a quadratic term, and sex of the individuals (as provided by the caregivers at the zoos or the person in charge of organizing the sample collection and known by direct phenotypic observation when

existent sexual dimorphism allows it or by molecular sexing with PCR when not), as predictive variables of glucose and glycation levels. Relative age was defined as its proportion to maximum lifespan of the species (i.e. divided by it), and logit transformation was performed for this ratio to avoid the particular distribution characteristics of proportion data, as Gaussian distribution was assumed in the models. We included glucose as a covariate for the glycation models, and body mass in every case, when it was available for the individuals. These models included a total of 239 individuals from 49 species (see *Appendix 1—table 1*).

We performed a model (MCMCglmm with species as a random factor) comparing the number of lysines exposed in the albumin of each species with the glycation levels measured in our study. This analysis aimed to assess whether this factor could represent a meaningful mechanism of resistance to glycation, as outlined in *Anthony-Regnitz et al., 2020*. When the albumin amino acids sequence for a species we measured was not available on NCBI or UniprotKB (the sources used to determine the number of lysines of each species' albumin; see below for additional details), we selected a closely related species, typically from the same genus (see *Appendix 1—table 2* for a list of the species selected and their correspondence to ours). In total, 19 species were included in this analysis. Finally, the models on glucose and glycation values controlling for the bird orders included in the dataset (see *Supplementary file 5*) were also performed as MCMCglmm, with two models considering the species averages (one for glucose and one for glycation) and two including the intraspecific values and therefore the species as a random factor to control for pseudo-replication due to multiple sampling of the same species. The species were allocated to the same orders employed on the classification given by the most updated version of Birds of the World (*Billerman et al., 2022*). The orders were organized in the model by alphabetic order, Accipitriformes constituting thus the intercept of all of them.

## Quantification of exposed lysine residues

The aminoacidic sequences of albumins were searched in UniprotKB and the NCBI database. For 20 species, the sequences were submitted to the PHYRE2 server to generate a PDB structure. Then, using DEPTH, the accessible surface area (ASA) was calculated for each residue in the sequences. We used a custom Python script to determine whether each lysine residue is exposed or not. To achieve this, the ASA of the lysine side chain was divided by 205 angstroms (the maximum observed accessible surface area for a lysine; *Jia et al., 2016*; *Xie et al., 2020*). If it exceeded 0.25, the residue was considered exposed. This methodology allowed us to determine the count of exposed lysine residues in our avian albumin sequences.

PHYRE2: http://www.sbg.bio.ic.ac.uk/~phyre2/html/page.cgi?id=index
DEPTH: https://bio.tools/depth ; http://mspc.bii.a-star.edu.sg/depth (Depth web server is actually dead).

**Appendix 1—table 1.** Main set of models performed with the number of species and individuals included in them.

A total of 10 MCMCglmm models were performed, having either plasma glucose or albumin glycation levels as response variable and a different number of species and total datapoints depending on if the life history (LH) traits were included (i.e. maximum lifespan, clutch mass, and developmental time) or only diet and body mass (also glucose in the glycation models). The models considering age and 201 sex did not include life history traits nor diet.

| | Species averages | | | | Intraspecific variation | | | | Age and sex | |
| | Glucose | | Glycation | | Glucose | | Glycation | | Glucose | Glycation |
| **Models performed and N for each** | Non-LH | LH | Non-LH | LH | Non- LH | LH | Non- LH | LH | | |
| Number of species | 88 | 66 | 88 | 66 | 75 | 58 | 75 | 58 | 49 | 49 |

*Appendix 1—table 1 Continued on next page*

*Appendix 1—table 1 Continued*

| | Species averages | | | | Intraspecific variation | | | | Age and sex | |
|---|---|---|---|---|---|---|---|---|---|---|
| | Glucose | | Glycation | | Glucose | | Glycation | | Glucose | Glycation |
| **Models performed and N for each** | Non-LH | LH | Non-LH | LH | Non-LH | LH | Non-LH | LH | ☒ | ☒ |
| Number of individuals | ☒ | ☒ | ☒ | ☒ | 379 | 316 | 379 | 316 | 239 | 239 |

**Appendix 1—table 2.** List of species used in the models testing the relationship between the number of albumin's exposed lysines and its glycation rates.
On the left, the list of considered species and on the right, the list of species used as references for the number of exposed lysines. The ones in bold are those that coincide.

| Species in our dataset | Original species |
|---|---|
| *Anas platyrhynchos* | *Anas platyrhynchos* |
| Anser anser | Anser brachyrhynchus* |
| Anser indicus | Anser cygnoides* |
| *Aptenodytes patagonicus* | *Aptenodytes patagonicus* |
| Tachimarptos melba | Apus apus |
| Aythya baeri | Aythya fuligula |
| *Balearica regulorum* | *Balearica regulorum* |
| *Bubo bubo* | *Bubo bubo* |
| *Cariama cristata* | *Cariama cristata* |
| *Cygnus atratus* | *Cygnus atratus* |
| *Eudyptes chrysolophus* | *Eudyptes chrysolophus* |
| Limosa limosa | Limosa lapponica |
| *Numida meleagris* | *Numida meleagris* |
| Leucocarbo verrucosus | Phalacrocorax carbo |
| *Phoenicopterus ruber* | *Phoenicopterus ruber* |
| *Pygoscelis papua* | *Pygoscelis papua* |
| *Rhea pennata* | *Rhea pennata* |
| *Taeniopygia guttata* | *Taeniopygia guttata* |
| *Tauraco erythrolophus* | *Tauraco erythrolophus* |
| Turdus merula | Turdus rufiventris |

*For these species, as they both belong to the same genus, an average was calculated for both the number of exposed lysines and the glycation levels, considering them in the analyses as one. This way, *Aser anser* and *Anser indicus* were substituted by an *Anser* sp. average glycation value, and *Anser brachyrhynchus* and *Anser cygnoides* by an average number of lysine exposed. The actual values were very similar in both cases, so this average is likely to reflect realistic values.

