## [Editor Report · eLife Assessment]

This **important** study uses extensive comparative analysis to examine the relationship between plasma glucose levels, albumin glycation levels, and diet and life history, within the framework of the ‘pace of life syndrome’ hypothesis. The evidence that glucose is positively correlated with glycation levels and lifespan is **convincing** and, although there are some limitations related to data collection, they likely make the statistically significant findings more conservative. As the first extensive comparative analysis of glycation rates, life history, and glucose levels in birds, the study has the potential to be of interest to evolutionary ecologists and the ageing research community more broadly.

---

## [Referee Report · Reviewer #2 (Public review)]

Summary:

In this extensive comparative study, Moreno-Borrallo and colleagues examine the relationships between plasma glucose levels, albumin glycation levels, diet and life-history traits across birds. Their results confirmed the expected positive relationship between plasma blood glucose level and albumin glycation rate but also provided findings that are somewhat surprising or contrast with findings of some previous studies (positive relationships between blood glucose and lifespan, or absent relationships between blood glucose and clutch mass or diet). This is the first extensive comparative analysis of glycation rates and their relationships to plasma glucose levels and life history traits in birds that is based on data collected in a single study, with blood glucose and glycation measured using unified analytical methods (except for blood glucose data for 13 species collected from a database).

Strengths:

This is an emerging topic gaining momentum in evolutionary physiology, which makes this study a timely, novel and important contribution. The study is based on a novel data set collected by the authors from 88 bird species (67 in captivity, 21 in the wild) of 22 orders, except for 13 species, for which data were collected from a database of veterinary and animal care records of zoo animals (ZIMS). This novel data set itself greatly contributes to the pool of available data on avian glycemia, as previous comparative studies either extracted data from various studies or a ZIMS database (therefore potentially containing much more noise due to different methodologies or other unstandardised factors), or only collected data from a single order, namely Passeriformes. The data further represents the first comparative avian data set on albumin glycation obtained using a unified methodology. The authors used LC-MS to determine glycation levels, which does not have problems with specificity and sensitivity that may occur with assays used in previous studies. The data analysis is thorough, and the conclusions are substantiated. Overall, this is an important study representing a substantial contribution to the emerging field evolutionary physiology focused on ecology and evolution of blood/plasma glucose levels and resistance to glycation.

Weaknesses:

Unfortunately, the authors did not record handling time (i.e., time elapsed between capture and blood sampling), which may be an important source of noise because handling-stress-induced increase in blood glucose has previously been reported. Moreover, the authors themselves demonstrate that handling stress increases variance in blood glucose levels. Both effects (elevated mean and variance) are evident in Figure ESM1.2. However, this likely makes their significant findings regarding glucose levels and their associations with lifespan or glycation rate more conservative, as highlighted by the authors.

---

## [Author Response]

The following is the authors’ response to the previous reviews

**Public Reviews:**

**Reviewer #2 (Public review):**
SummaryIn this extensive comparative study, Moreno-Borrallo and colleagues examine the relationships between plasma glucose levels, albumin glycation levels, diet and lifehistory traits across birds. Their results confirmed the expected positive relationship between plasma blood glucose level and albumin glycation rate but also provided findings that are somewhat surprising or contrast with findings of some previous studies (positive relationships between blood glucose and lifespan, or absent relationships between blood glucose and clutch mass or diet). This is the first extensive comparative analysis of glycation rates and their relationships to plasma glucose levels and life history traits in birds that is based on data collected in a single study, with blood glucose and glycation measured using unified analytical methods (except for blood glucose data for 13 species collected from a database).StrengthsThis is an emerging topic gaining momentum in evolutionary physiology, which makes this study a timely, novel and important contribution. The study is based on a novel data set collected by the authors from 88 bird species (67 in captivity, 21 in the wild) of 22 orders, except for 13 species, for which data were collected from a database of veterinary and animal care records of zoo animals (ZIMS). This novel data set itself greatly contributes to the pool of available data on avian glycemia, as previous comparative studies either extracted data from various studies or a ZIMS database (therefore potentially containing much more noise due to different methodologies or other unstandardised factors), or only collected data from a single order, namely Passeriformes. The data further represents the first comparative avian data set on albumin glycation obtained using a unified methodology. The authors used LC-MS to determine glycation levels, which does not have problems with specificity and sensitivity that may occur with assays used in previous studies. The data analysis is thorough, and the conclusions are substantiated. Overall, this is an important study representing a substantial contribution to the emerging field evolutionary physiology focused on ecology and evolution of blood/plasma glucose levels and resistance to glycation.WeaknessesUnfortunately, the authors did not record handling time (i.e., time elapsed between capture and blood sampling), which may be an important source of noise because handling-stress-induced increase in blood glucose has previously been reported. Moreover, the authors themselves demonstrate that handling stress increases variance in blood glucose levels. Both effects (elevated mean and variance) are evident in Figure ESM1.2. However, this likely makes their significant findings regarding glucose levels and their associations with lifespan or glycation rate more conservative, as highlighted by the authors.
**Recommendations for the authors:**

**Reviewer #2 (Recommendations for the authors):**
I understand that your main objective regarding glycation rate and lifespan, was to analyse the species resistance to glycation with respect to lifespan, while factoring out the species-specific variation in blood glucose level. However, I still believe that the absolute glycation level (i.e., not controlled for blood glucose level) may also be important for the evolution of lifespan. Given that blood glucose is positively related to both glycation and lifespan (although with a plateau in the latter case), lifespan could possibly be positively correlated with absolute glycation levels. If significant, that would be an interesting and counterintuitive finding, which would call for an explanation, thereby potentially stimulating further research. If not significant, it would show that long-lived species do not have higher glycation levels, despite having higher blood glucose levels, thereby strengthening your argument about higher resistance of longlived species to glycation. So, in my opinion, the inclusion of an additional model of glycation level on life-history traits, without controlling for blood glucose, is worth considering.

We include now this model as supplementary material, indicating it in several parts of the text, including some of these issues we discussed here.

Lines 230-231: Please, provide a citation for these GVIF thresholds

We include it now.

Figure 3: I think that showing both glucose and glycation rate on the linear scale, rather than log scale, would better illustrate your conclusion - the slowing rise of glycation rate with increasing glucose levels.

That is a good point, although it may also be confusing for readers to see a graph that represents the data in a different way as the models. Maybe showing both graphs (as 3.A and 3.B) can solve it?

Figure 4. I recommend stating in the caption that the whiskers do not represent interquartile ranges (a standard option in box plots) but credible intervals as mentioned in the current version of the public author response.

Sorry about that, it was missed. Now it is included. Nevertheless, interquartile ranges from the posterior distributions can still be observed here represented with the boxes. Then the whiskers are the credible intervals.